# A Multi-Stage Approach for Cardiovascular Risk Assessment from Retinal Images Using an Amalgamation of Deep Learning and Computer Vision Techniques

**DOI:** 10.3390/diagnostics14090928

**Published:** 2024-04-29

**Authors:** Deepthi K. Prasad, Madhura Prakash Manjunath, Meghna S. Kulkarni, Spoorthi Kullambettu, Venkatakrishnan Srinivasan, Madhulika Chakravarthi, Anusha Ramesh

**Affiliations:** 1Research and Development, Image Processing and Analysis, Forus Health Private Ltd., Bengaluru 560070, India; madhuraprakash@forushealth.com (M.P.M.); meghnask@forushealth.com (M.S.K.); spoorthi.k@forushealth.com (S.K.); venkat@forushealth.com (V.S.); 2Department of Cardiology, Apollo Hospitals, Bengaluru 560076, India; chakravarthimadhulika@gmail.com; 3Department of OBGyn, St. John’s Medical College, Bengaluru 560034, India; anusha.ramsh@gmail.com

**Keywords:** cardiovascular diseases, fundus images, retinal vascular parameters, artificial intelligence, computer vision, multi-stage architecture

## Abstract

Cardiovascular diseases (CVDs) are a leading cause of mortality worldwide. Early detection and effective risk assessment are crucial for implementing preventive measures and improving patient outcomes for CVDs. This work presents a novel approach to CVD risk assessment using fundus images, leveraging the inherent connection between retinal microvascular changes and systemic vascular health. This study aims to develop a predictive model for the early detection of CVDs by evaluating retinal vascular parameters. This methodology integrates both handcrafted features derived through mathematical computation and retinal vascular patterns extracted by artificial intelligence (AI) models. By combining these approaches, we seek to enhance the accuracy and reliability of CVD risk prediction in individuals. The methodology integrates state-of-the-art computer vision algorithms and AI techniques in a multi-stage architecture to extract relevant features from retinal fundus images. These features encompass a range of vascular parameters, including vessel caliber, tortuosity, and branching patterns. Additionally, a deep learning (DL)-based binary classification model is incorporated to enhance predictive accuracy. A dataset comprising fundus images and comprehensive metadata from the clinical trials conducted is utilized for training and validation. The proposed approach demonstrates promising results in the early prediction of CVD risk factors. The interpretability of the approach is enhanced through visualization techniques that highlight the regions of interest within the fundus images that are contributing to the risk predictions. Furthermore, the validation conducted in the clinical trials and the performance analysis of the proposed approach shows the potential to provide early and accurate predictions. The proposed system not only aids in risk stratification but also serves as a valuable tool for identifying vascular abnormalities that may precede overt cardiovascular events. The approach has achieved an accuracy of 85% and the findings of this study underscore the feasibility and efficacy of leveraging fundus images for cardiovascular risk assessment. As a non-invasive and cost-effective modality, fundus image analysis presents a scalable solution for population-wide screening programs. This research contributes to the evolving landscape of precision medicine by providing an innovative tool for proactive cardiovascular health management. Future work will focus on refining the solution’s robustness, exploring additional risk factors, and validating its performance in additional and diverse clinical settings.

## 1. Introduction

Cardiovascular diseases (CVDs) are a significant global health burden and stand as a predominant global cause of mortality [1]. Factors such as sedentary lifestyles, unhealthy diets, tobacco use, and an aging population contribute to the increasing incidence of CVDs worldwide. According to the World Health Organization (WHO), CVDs account for most non-communicable disease-related deaths globally, making them a leading cause of mortality. In 2019, an estimated 18 million individuals succumbed to CVDs, accounting for 32% of all global deaths according to the World Health Organization. Notably, stroke and ischemic heart disease alone caused 15.2 million deaths in 2015, constituting 85.1% of total cardiovascular-related fatalities [2]. Many CVDs are preventable by addressing various risk factors, including individual factors like age, gender, smoking habits, blood pressure, body mass index (BMI), and metabolic indicators such as glucose and cholesterol levels [3]. The escalating prevalence of CVDs, encompassing a spectrum of conditions affecting the heart and blood vessels, warrants a closer examination of their impact both on a global scale and specifically within India.

In India, the shifting epidemiological landscape, fueled by swift urbanization, changing lifestyles, and a demographic shift toward an aging population, has led to a notable increase in CVDs. Studies also indicate that CVDs account for nearly 28% of all deaths in the country, making them the leading cause of mortality. Projections indicate that the prevalence of CVDs is on the rise, with estimates anticipating a surge of more than 50% in CVD-related fatalities by 2030. The escalating prevalence and impact of CVDs both globally and in India underscore the urgent need for comprehensive strategies aimed at prevention, early detection, and effective management.

The timely detection of CVDs plays a pivotal role in enabling effective clinical interventions. Incorporating a combination of computer vision approaches like image processing, machine learning, and AI into the diagnostic process holds promise for enhancing early detection for the effective assessment of CVD conditions. Early detection of cardiovascular risk factors and diseases is paramount in curbing the escalating rates of morbidity and mortality. Early risk detection empowers individuals to make lifestyle modifications, initiate appropriate medications, and adopt preventive measures to mitigate their risk of developing CVDs. Advancements in medical technology have introduced innovative tools and diagnostic techniques that aid in early risk detection. Non-invasive methods, such as retinal assessment, and other imaging modalities, such as ultrasound or cardiac magnetic resonance imaging (MRI), empower healthcare professionals to conduct a thorough and proactive evaluation of cardiovascular health.

The retina is a light-sensitive layer of tissue located at the back of the eye that serves as a noninvasive “window” to provide insights into the cardiovascular health of an individual. The retina mirrors the anatomical structure and functions of the cardiac vasculature, and any alterations in systemic circulation can be observed in this tissue. Studies have linked various retinal characteristics to the risk of CVD, encompassing factors related to retinal vascular geometry such as vessel caliber ratios (arteriovenous ratio), branching angle, tortuosity, and fractal dimension. Vascular network patterns and specific retinal pathologies like cotton wool spots, arteriovenous nicking, and microaneurysms are also linked to possible risk for CVD [4,5,6,7]. Retinal findings, such as alterations in vessel structure or the presence of specific lesions, can predict the risk of future cardiovascular events like stroke, heart attack, or other systemic vascular diseases. This predictive potential highlights the importance of utilizing retinal assessments as a non-invasive screening tool for identifying individuals at higher risk for cardiovascular issues.

Retina-based cardiovascular assessment involves examining the blood vessels in the retina to gain insights into an individual’s cardiovascular health. Advanced imaging techniques, such as fundus photography, are used to capture high-resolution images of the retinal vessels. These images are then processed and analyzed for the computation of various retinal parameters and presence of abnormalities such as microaneurysms, hemorrhages, or exudates. Retina-based cardiovascular assessment holds promise in clinical practice as a complementary method for evaluating cardiovascular health. It enables early detection and monitoring of cardiovascular risk factors, aiding in risk stratification and guiding preventive interventions. Integrating retinal examination into routine health screenings could enhance cardiovascular risk assessment and facilitate timely interventions to mitigate potential health risks.

The substantial correlation observed between CVD risk and various retinal parameters underscores the importance of investigating existing systems [8,9,10,11,12,13,14,15,16] In a study by Cheung et al. [17], which followed 4593 patients without established CVD, it was found that LV concentric hypertrophy was independently associated with narrower retinal arteriolar caliber and the presence of retinopathy, even after adjusting for demographics and traditional CVD risk factors. Decreased central retinal arteriolar equivalent (CRAE) and increased central retinal venular equivalent (CRVE) were associated with echocardiographic measures of both LV systolic and diastolic dysfunction, as demonstrated by Chandra et al. [18].

Arnould L et al. [7] investigated the association between the retinal vascular network, assessed using the Singapore “I” Vessel Assessment (SIVA) software version 3.0, and cardiovascular history and risk factors in elderly individuals, as part of the Montrachet study, a population-based study. Fathalla et al. [19] aimed to identify patterns and associations between retinal vascular features and CVD risk. Rudnicka et al. [20] performed a AI-enabled retinal vasculometry anaysis. Seidelmann [6] investigated the association between retinal vessel calibers and long-term CVD outcomes, assessed using Cox proportional hazards regression analysis. Muhammad et al., 2015 [21], performed quantitative measurements of retinal vessel parameters, used to identify abnormalities in vessel structure and to track changes over time in response to treatment or disease progression. Poplin et al. [22] investigated the feasibility of using DL algorithms to predict cardiovascular risk factors. Pinto et al. [23] made use of retinal scans and minimal personal information in order to predict myocardial infarction. Fetit et al. [13] incorporated a multimodal approach to cardiovascular risk stratification in patients with type 2 diabetes. Peng et al. [24] used a combination of DL techniques and traditional machine learning methods to extract relevant features from retinal images and integrate them with clinical data and demographic information. Lee et al. [25] made use of DL techniques in retinal imaging for predicting CVD events in individuals with prediabetes and diabetes. Zekavat al., 2022 [26], applied DL methods for analyzing complex phenotypes and integrating retinal imaging data with genetic information.

The literature indicates the potential of the retina to reveal indicators of cardiovascular health, and systems that compute and analyze the retinal parameters exist. However, study of the approaches of the existing systems indicates that there is the need for an advanced and holistic approach that incorporates the latest advancements in image analysis and AI to provide a more accurate and comprehensive assessment of CVD risk. This study aims to address this gap by proposing a novel hybrid approach that combines computer vision techniques with AI-driven retinal vascular pattern analysis. This hybrid architecture enhances the robustness and efficacy of CVD risk prediction by leveraging both handcrafted features and advanced DL algorithms.

This study incorporates a hybrid and multi-stage methodology, which includes an amalgamation of DL and computer vision techniques to extract and analyze the retinal parameters for cardiovascular risk assessment. By integrating hand-crafted feature-based retinal parameter computation and off-the-shelf feature extraction techniques into a comprehensive framework, our study offers a tailored solution for CVD risk evaluation that goes beyond existing approaches. Additionally, our methodology has been validated through meticulous clinical trials and has achieved an accuracy of 85% in the hybrid approach. The novelty of our study lies in the development of a customized architecture that combines multiple methodologies to provide a more precise assessment of cardiovascular health. This innovative approach has the potential to significantly advance the field of automated CVD diagnosis derived from retinal fundus imaging. The proposed methodology offers a significant advancement in non-invasive CVD risk assessment because its execution necessitates solely the acquisition of retinal images from the subject under evaluation, without requiring additional demographic or clinical data. This approach enhances accessibility and scalability, potentially revolutionizing CVD risk stratification in clinical practice.

## 2. Materials and Methods

This work aims to achieve an automated CVD diagnosis derived from retinal fundus imaging (RFI). The fundus images were collected from patients at clinical trials for this purpose. A comprehensive set of metadata associated with the patients, like the specific CVD condition being diagnosed and treated and the medications prescribed, were used as a reference to segregate and curate the data into cardio risk and non-cardio risk categories. A multi-stage architecture based on a hybrid approach incorporating computer vision and AI techniques was used for CVD risk prediction in validation. Four exhaustively trained and validated AI models were assimilated in this multistage architecture. The first one’s function is to assess the quality and validity of the fundus image captured, namely the “outlier detection model”. The second one is an “optic disc (OD) location model” to segment the OD in the retina. The third is a joint segmentation and classification model to segment the blood vessels in the retina and classify them as artery and vein. The fourth in this pipeline is a customized complementary DL model based on transfer learning for the binary classification of fundus images as risk for cardio or no-risk. The developed solution was validated in a clinical trial setting to establish its accuracy. The DL-based classification solution resulted in accuracy, sensitivity, specificity, and F1 scores of 87.5, 85.29, 86.36, and 86.15, respectively. The combined approach achieved an overall accuracy of 85% in clinical validation.

### 2.1. Research Design

This study adopted a cross-sectional design, collecting data from a cohort of participants. Selection of the participants involved individuals aged 20–80 in the two categories. The first category comprised cases from individuals diagnosed with various CVDs such as ischemic heart disease (IHD), angina, coronary artery bypass graft (CABG), atrial flutter, concentric left ventricular hypertrophy (LVH), cardiomyopathy, and reduced left ventricular (LV) function, along with individuals with hypertension. Subjects in the hypertension category that presented with other factors like high cholesterol levels, the presence of diabetes, and age greater than 65 were also included in the first category in order to develop a solution that can act as an early biomarker for the detection of cardio risk. The second category consisted of data from healthy controls. These individuals did not have any concurrent cardiovascular or neurological conditions such as hypertension, diabetes, stroke, or a history of eye trauma. They served as a comparison group to assess the differences in retinal fundus imaging findings between those with cardiovascular diseases and those without such conditions. The study was conducted at Ramesh Cardiology clinic [27] in collaboration with clinical experts. The study obtained informed consent from all participants after providing information about the study. The study ensured strict confidentiality protocols for participant data storage and handling. The study guaranteed participant safety throughout the imaging procedures. The study focused on an age group from 20 to 80 with two categories. The clinical trial resulted in the collection of a sample size that included 379 cases in the first category and 150 cases in the second category.

This study utilized a non-invasive retinal imaging technique for capturing fundus images depicting the details of the retina. Cardiovascular health data, including blood pressure, lipid profile, BMI, and relevant medical history through standardized clinical assessments were collected as metadata for each participant. These metadata served as a deciding factor for categorizing and curating the collected data on which the experimentation was conducted.

### 2.2. Implementation Approach

This research aims to explore the potential correlations between retinal imaging and cardiovascular health indicators obtained from the diagnosis information for the subjects. This study proposes a methodology that integrates retinal imaging techniques with established cardiovascular assessment parameters to explore potential diagnostic and predictive associations using a multistage architecture based on a hybrid approach. One stage is related to retinal-parameter-based analysis and the other one uses classification based on DL techniques.

The retina-based cardiovascular risk assessment system encompasses a sophisticated amalgamation of medical imaging, DL algorithms, and cardiovascular risk prediction models. The following are the modules of the proposed system. The retinal image is captured using a fundus camera. For our study, the images were captured from 3nethra Classic HD device, developed by Forus Health Pvt. Ltd., Bengaluru, India [28]. The images were captured in non-mydriatic, optic disc centric mode.

The next step incorporates the development of a DL classification model, using retinal images to predict and classify cardiovascular health indicators. It integrates advanced neural network architecture with labeled retinal image datasets to enable accurate classification.

The high-level architecture of the proposed approach is depicted in Figure 1. The input to the CVD assessment system is a retinal fundus image. This is followed by the outlier detection module, which discards any other image other than a fundus image. The proposed method mandates the image to be in optic centric mode since the analysis is based on an optic disc centric approach, where the regions around the optic disc are considered prevalent for systemic analysis [29]. The optic disc detection module employs a U-net segmentation model [30] for identification of the optic disc region. Any fundus image that does not have an optic disc is discarded by this module. Once a valid image is fed to the system, the approach is to analyze the image from two different perspectives, using parameter-based analysis and image-based classification employing DL techniques. Parameter-based analysis utilizes image processing and DL methods, analyzes the vascular structures of the retina, and computes the parameters associated with the blood vessels.

The method incorporates the extraction of relevant parameters by analyzing the blood vessels around the optic disc in zones. The blood vessel segmentation and artery vein classification module adopts a customized architecture based on a modified U-net model to segment vessels from the background and to predict the type of vessel (artery/vein). Retinal parameters like vessel morphology, vessel caliber, retinal vascular fractal dimension, and arteriovenous ratio (AVR) that are relevant for CVD conditions are identified from various literature sources and correlated with the clinical trial data and studies performed at the clinical trials. Out of these, a subset of the parameters is found to correlate with the subject’s diagnostic data. Risk for CVD is assessed based on the reference range of the parameters obtained from the literature, correlated with the diagnostic information obtained from the clinical study. The risk for the image is identified based on the ranges that were determined and flagged as “*cardio suspect*” if it falls within the range, and otherwise it is flagged as “*non-cardio suspect*”.

The latter method employs a DL classification model to develop a training model to classify a given retinal image as “*cardio suspect*” or “*non-cardio suspect*”. This approach utilizes EfficientNetB0 [31], which is a convolutional neural network (CNN) trained on more than million images from the ImageNet database [32]. The threshold for classification is decided based on the distribution of the prediction values for both classes (“*cardio suspect*” or “*non-cardio suspect*”).

The work aims at the combined assessment of CVD risk using the outcome of both the approaches described above to improve the overall performance and to increase the confidence of the risk assessment.

The inference is strengthened by the amalgamation of the outcome of the approaches mentioned above. The outcome is “cardio suspect” if it yields “cardio suspect” from the outcome of both the parameter-based and DL-based classification methods. This decision is made to ensure that the subject’s status as a cardio suspect is confirmed through a double verification process, thereby minimizing the possibility of subjecting individuals to unnecessary stress and incorrect diagnoses or treatments. The specifics of both approaches are outlined in the following two sub-sections.

#### 2.2.1. Parameter-Based Approach

The outlier detection model precedes both approaches. The architecture of this model is based on a sequential layered convolution neural network (CNN). The model is trained using an Adam optimizer with learning rate 1 × 10^−4^. The intent of this model is to filter out all the images under the non-fundus category and any poor-quality images without retinal structure details. This model was trained, tested, and validated on the Forus Health [28] internal images. Table 1 lists the training, test, and validation dataset details of this model. This model individually has achieved a validation accuracy of 98 percent.

The optic disc (OD) segmentation phase is next in the sequence in the parameter-based approach. The OD detection model is based on UNet architecture. The U-net model consists of a contracting path and an expansive path. The contracting path consists of the repeated application of (3 × 3) convolutions, each followed by a batch normalization layer and a rectified linear unit (ReLU) activation and dropout and a (2 × 2) max pooling operation with stride 2 for down sampling. At each down sampling phase, the model doubles the number of feature channels. The contracting path captures the context of the input image to do segmentation. The expansive path consists of an up sampling of the feature map followed by a (2 × 2) convolution (up-convolution) that halves the number of feature channels, a concatenation with the corresponding feature map from the contracting path, and two (3 × 3) convolutions, each followed by batch norm, dropout, and a ReLU. The purpose of this expanding path is to enable precise localization combined with contextual information from the contracting path. At the final layer, a (1 × 1) convolution is used to map each component feature vector to the desired number of classes. The model was trained after augmenting 900 original fundus images and their corresponding ground truth OD masks. The model achieved an intersection over union (IoU) of 0.94.

The subsequent step in the sequence involves a joint blood vessel segmentation and artery–vein classification model. This model is essential as a preliminary step to compute all the retinal vascular parameters accurately. This step processes the input image sequentially, allowing it to capture both local and global information effectively. This sequential processing helps in accurately segmenting blood vessels and classifying arteries and veins. For blood vessel segmentation, the proposed model employs a technique called semantic segmentation, where each pixel in the image is classified as either belonging to a blood vessel or background. This segmentation process helps in identifying the location and extent of blood vessels in the fundus image. After segmenting the BVs, the model classifies them into arteries and veins using classification techniques. This classification is based on features learned by the network during training, such as vessel width, branching patterns, and intensity profiles. The model was trained after augmenting 500 original fundus images and their corresponding ground truth segmentation and classification masks. The model achieved an intersection over union (IoU) of 0.91.

A sample zone-wise division of retinal fundus images and the segmented–classified artery–veins (AV) on the fundus image are shown in Figure 2. In the zonal depiction of the retinal fundus image, the region from the optic disc margin to 0.5 optic disc diameters (ODD) is coined as Zone A, Zone B refers to 0.5 to 1 ODD, and Zone C is described as the region which is 0.5 to 2.0 ODD away from the margin of the optic disc [33]. The top 6 arteries (A1–A6) and veins (V1–V6) are identified in all the zones and various parameters are computed from them. Once the retinal blood vessels are segmented and classified, the top 6 arteries (A1–A6) and veins (V1–V6) marked in red and blue in Figure 2 are ranked and selected based on criteria such as vessel length, intensity, prominence and connectivity.

Specific parameters and features in the retina that correlate with cardiovascular health like vessel caliber, tortuosity, and branching patterns are computed and analyzed. Changes in vessel caliber, such as arteriolar narrowing or venular dilation, have been associated with hypertension and other cardiovascular risk factors. Increased tortuosity and branching irregularities may indicate alterations in blood flow dynamics, potentially linked to atherosclerosis or vascular remodeling. By analyzing these retinal vascular characteristics, we can glean valuable information about an individual’s cardiovascular health, enabling early detection for the risk of cardiovascular events.

In this study, a comprehensive analysis of retinal vasculature was conducted. We calculated seven different parameters for both retinal arteries and veins. Utilizing advanced image analysis techniques, parameters specific to arterial and venous structures were calculated. This method helps to thoroughly assess the distinct traits of both arterial and venous networks in the retina. Parameters computed for retinal arteries and veins typically involve specific characteristics that are assessed separately for each type of vessel. The parameters computed for retinal arteries and venules are:Branching coefficientJunction exponentMean artery/vein widthOptimality ratioPath lengthSimple tortuosityVessel diameter reduction

Additionally, we identified common parameters between arteries and veins:CRAECRVEArteriovenous ratioFractal dimension

Branching Coefficient:

This refers to a quantitative measure used to describe the branching pattern or complexity of the vessels within the retinal vasculature network. Given the parent branch width $r_{0}$ and daughter vessel widths $r_{1\;},\;r_{2}$, etc., then the branching coefficient (BR) is calculated using Equation (1) as:
$r_{0}$—Store parent vessel width$r_{i}$—Store first and second daughter branch vessel width. i=1, 2 Calculate $r_{0}^{2}$ and squares of all $r_{i}$
(1)$$Branching\;Coefficient\;=\;\frac{r_{1}^{2}+r_{2}^{2}}{r_{0}^{2}}$$

Junction Exponent

The junction exponent is a measure that describes the relationship between the number of branches and the order of branching in retinal morphology. Given the parent branch width $r_{0}$ and daughter vessel widths $r_{1},\;r_{2},\;r_{3},$ etc., then the junction exponent (JE) is calculated using Equation (2) as:
$r_{0}$—Store parent vessel width$r_{i}$—Store ith daughter branch vessel width i = 1 to *n*

(2)$$JE=r_{0}^{3}-\left(r_{1}^{3}+r_{2}^{3}+r_{3}^{3}\right)$$*JE*—Junction exponent

Mean Artery/Vein Width

The mean artery/vein vessel width refers to the average diameter or width of retinal arteries or veins within the eye’s vasculature. It is calculated using Equation (3) as:Consider a vessel centerline pixel for selecting a cross-section.Search an edge pixel from a certain distance (r) and angle (a).Each time an edge pixel is found, obtain the opposite edge pixel by adding 180° to (a) and varying the distance (r) from the centerline pixel.Calculate the width by measuring the minimum Euclidean distance between two opposite edge points.
(3)$$MW\;=\sqrt{\left({\left(x_{1}-x_{2}\right)}^{2}+{\left(y_{1}-y_{2}\right)}^{2}\right)}$$
*MW*—Mean Width

Optimality Ratio

Given the parent branch width $r_{0}$ and daughter vessel widths $r_{1},r_{2},$ etc., then the optimality ratio (OR) is calculated using Equation (4) as:
$r_{0}$—Store parent vessel width$r_{i}$—Store first and second daughter branch vessel width. i=1, 2Calculate $r_{0}^{3}$ and cubes of all $r_{i}$
(4)$$x=\frac{\left(\sum_{i=1}^{2}r_{i}^{3}\right)}{\left(2*r_{0}^{3}\right)}$$Finally, calculate OR as OR = $\sqrt[{3}]{x}$

Path Length

The path length of a vessel refers to the distance or length along the course of a blood vessel.
Get a stable retinal vascular tree.Find the single largest external path length in the tree.Find the total sum of external path lengths in the tree.Find the total number of exterior–interior path lengths in the tree.

Simple Tortuosity

Simple tortuosity is the degree of bending observed along the course of a blood vessel, without considering complex irregularities or intricate patterns in its shape. It can be estimated as the difference between the actual path length of the vessel segment and the straight-line length of the segment divided by the straight-line length.

Vessel Diameter Reduction

Vessel diameter reduction refers to the decrease in the diameter or width of a blood vessel compared to its normal size. Given the parent and daughter vessel diameters, vessel diameter reduction can be calculated as the deviation of vessels from the normal diameter measure.
$r_{0}$—Store parent vessel width$r_{i}$—Store first and second daughter branch vessel width. i=1, 2 d—Deviation of vessels from the normal diameter measure

CRAE and CRVE

These are summaries of the measurements of vascular equivalent caliber representing the equivalent single vessel parent width of the six largest arterioles and venules, based on the Knudtson–Parr–Hubbard formula [34]. The largest trunk is paired with the smallest one and so on till only one remains.
Calculate widths of all veins and arterioles.Rank them in decreasing order.Select the top 6 veins and arteries.Calculate the CRAE and CRVE using the following formulae given in Equations (5) and (6):
(5)$$CRAE\;=\;0.88\;*\;{\left(w_{1}^{2}\;+\;w_{2}^{2}\right)}^{\frac{1}{2}}$$
(6)$$CRVE\;=\;0.95\;*\;{\left(w_{1}^{2}\;+\;w_{2}^{2}\right)}^{\frac{1}{2}}$$
$w_{1}$—width of narrower branch$w_{2}$—width of wider branch
5.Repeat step 4 till a single value remains for both CRVE and CRAE.

The Knudtson–Parr–Hubbard formula is a widely used method to compute CRAE, a measure of retinal arteriolar caliber. The importance of using this formula lies in its ability to provide a standardized and reproducible way of quantifying retinal vessel caliber.

Arteriovenous Ratio

The ratio of the central retinal arteriolar equivalent to the central retinal venular equivalent is known as the atrioventricular ratio.
Find CRAE and CRVE.Use the formula given in Equation (7) to calculate AVR:
(7)$$AV\;Ratio\;=\;\frac{CRAE}{CRVE}$$

Fractal Dimension

Fractal dimension is a measure used to quantify the complexity or self-similarity of vessels.
Cover image by a sequence of grids of descending sizes.Record:The number of square boxes intersected by the image, *N*(*s*).The side length of the squares, *s*.Plot $\log\left(N\left(s\right)\right)\;\;\;\;\;\;\;\;\;\;\;\;\;\;\;\;\;\;\;\;\;\;\;\;\;\;\;\;\;\;\;\;\;\;\;\log\left(\frac{1}{s}\right)$.Get the regression slope D.Find the fractal dimension from Equation (8) as:
(8)$$\log\left(N\left(s\right)\right)=\log\left(K\right)+\;D\;\log\left(\frac{1}{s}\right)\;$$
K—Constant
$$N\left(s\right)\;\alpha \;\frac{1}{s^{-D}}$$

Among all the retinal parameters computed, three specific parameters—CRAE, AV ratio, and mean artery width—have been identified as differentiators for identifying cardiovascular disease from the clinical trials conducted. The importance of CRAE in cardiovascular disease detection from retinal images lies in its potential role as a non-invasive biomarker. The retinal vessels reflect the state of the microvasculature throughout the body. Alterations in retinal vessel caliber, such as a narrower arterioles, may suggest changes in the systemic circulation. Reduced CRAE has been associated with hypertension, a major risk factor for cardiovascular disease. Studies have suggested that changes in retinal vessel caliber, including CRAE, may be predictive of future cardiovascular events. Monitoring these changes over time can aid in early detection and intervention.

The AVR, also known as the arteriolar-to-venular ratio, is another parameter derived from retinal images that is significant in cardiovascular disease detection. The AVR is the ratio of the width of retinal arterioles to venules and is used to assess the health of the microcirculation. The AVR provides information about the relative sizes of retinal arterioles and venules. Changes in this ratio can indicate alterations in microvascular health, reflecting the state of the systemic circulation.

Deviations from a normal AVR have been associated with hypertension and an increased risk of cardiovascular events. A lower AVR (narrower arterioles relative to venules) is often linked to hypertension and is considered a potential indicator of cardiovascular risk. The width of retinal arterioles is also an important parameter in the analysis of retinal images for cardiovascular disease detection. Retinal arterioles are part of the microvascular network, and alterations in their width can reflect changes in the small blood vessels throughout the body. These changes may be associated with cardiovascular diseases or risk factors.

Narrowing of retinal arterioles has been associated with hypertension, a major risk factor for cardiovascular diseases. These parameters exhibit significant associations and provide significant distinguishing power, separating individuals affected by cardiovascular conditions from those without such health concerns. Observations in patients with cardiovascular disease include retinal arteriolar narrowing, decreased arteriovenous ratio, and reduced arteriole width. This approach of assessing cardiovascular health through retinal images represents a non-invasive and potentially early method for identifying individuals at risk of or affected by cardiovascular diseases.

#### 2.2.2. AI-Based CVD Analysis

Classification of the retinal images into cardio and non-cardio categories was implemented using a DL-based binary classifier to complement the results obtained using the zone-based analysis. The idea was to extract the off-the-shelf features and patterns from the fundus images using trained models that aid in distinguishing between cardio and non-cardio cases. To achieve this, a customized neural network architecture based on the transfer learning approach with standard EfficientNetB0 as the baseline model is utilized. Here, a pre-trained neural network model is used as a baseline to design a customized DL architecture. This architecture is used to train a model that can classify retinal images into two distinct classes, namely risk or no-risk for cardio. Transfer learning is employed here to leverage the pre-trained weights of the EfficientNetB0 architecture. However, the weights in the baseline model were not frozen. This approach was chosen to enhance the model’s ability to perform well on the specific binary classification task at hand. EfficientNetB0 [35] is a convolutional neural network architecture, and it is chosen for its efficiency in balancing model size and performance. The main idea behind EfficientNet is to achieve better performance by carefully balancing the model’s depth, width, and resolution. Here are some key characteristics about the EfficientNetB0 architecture:Depth-wise Separable Convolution:

EfficientNetB0 uses depth-wise separable convolutions, which consist of a depth-wise convolution followed by a point-wise convolution. This type of convolution reduces the number of parameters and computations compared to traditional convolutions, making the network more efficient.

Inverted Residual Blocks:

The architecture includes inverted residual blocks, inspired by MobileNetV2. Inverted residuals use shortcut connections and linear bottlenecks to improve the flow of information through the network.

Global Average Pooling (GAP):

Instead of using fully connected layers at the end of the network, EfficientNetB0 employs global average pooling. This helps reduce the number of parameters and avoids overfitting.

Scaling Coefficients:

EfficientNet introduces compound scaling to balance the network’s depth (number of layers), width (number of channels), and resolution (input image size). The scaling coefficients (phi) are used to control these factors, and for EfficientNetB0, phi is set to 1.

Input Image Size:

The default input image size for EfficientNetB0 is 224 × 224 pixels.

Number of Parameters:

The number of parameters in EfficientNetB0 is relatively small compared to other architectures, making it more efficient for deployment on various devices.

EfficientNetB0 offers several advantages over other models like efficiency across parameters, lower computational cost, faster training and inference, memory efficiency, easier deployment, fine-grained image recognition, and transfer learning. The binary classification process involves the following steps:Data Preprocessing:

Import the retinal images dataset.Resize images to match the input size expected by EfficientNetB0 i.e., (224, 224).Normalize pixel values to ensure consistent input to the model.The black border from the retinal images is removed to remove the unnecessary portion of the fundus image not required for training.

Architecture Customization:

Append additional layers to the base model.A batch normalization layer, drop out layer, regularization layer, and a fully connected dense layer are appended to the base model.

Transfer Learning:

Load the pre-trained EfficientNetB0 model, which has already learned features from a large dataset.Use the pre-trained weights from the initial layers of the network to retain the learned features.Customize the final layers to adapt the model for binary classification.

Training:

Split the dataset into training and validation sets.Train the model on the training set, fine-tuning the final layers for the specific task.Validate the model on the validation set to monitor performance.

Evaluation:

Assess the model’s performance using metrics such as accuracy, precision, recall, and F1 score.Use a separate test set to evaluate the model’s generalization to unseen data.

This approach harnesses the power of EfficientNetB0’s pre-trained features, enabling the model to effectively classify retinal images while minimizing the need for extensive training. The customized binary classification model summary details can be found in the Appendix A. The architecture was implemented utilizing the Python programming language, leveraging functionalities provided by the OpenCV and Keras libraries. The model was trained using the Adam (1 × 10^−3^) Optimizer and using binary cross-entropy as a loss function. The initial learning rate (LR) of training the model was kept at 0.00001. To ensure the smooth and faster convergence of the model, callback functions like early-stopping and reducing LR on plateau were included. The details of the training, testing, and validation dataset division are given below in Table 2. These images were augmented using the ImageDataGenerator functionality from the Keras library. Flipping, rotation, and brightness adjustments were incorporated and both categories were balanced to 1500 images using this functionality.

The sequence of pre-processing techniques applied to enhance the fundus images before passing these images to model training are given in Figure 3. The pre-processing steps are followed to enhance the vessel morphology before passing the image to the model. Firstly, based on the input image and the border mask, the black border is cropped and masked to ensure that the model does not look for any patterns in this irrelevant region on the image.

Next, a color channel splitting is performed on the masked image to extract the blue (B), green (G), and red (R) channels. Contrast Limited Adaptive Histogram Equalization (CLAHE) with a clip limit of 1.0 and a tile grid size of (25, 25) is then applied to enhance the intensity distribution of the green channel. Next, the intensity of the enhanced green channel is enhanced by multiplying it with a factor of 0.8. Finally, the original blue and red channels are merged with the updated green channel to create the final BGR image, which is then returned. This process enhances the image’s overall contrast and improves visual quality. Furthermore, gamma adjustment is performed on the contrast enhanced image. The inverse of the specified gamma value is used to create a lookup table that aids in adjusting pixel values. The details of the pre-processing steps are in the Appendix A.

## 3. Results

The clinical trials were conducted at Ramesh Cardiology Clinic, Bengaluru, India. The trials spanned 6 months from March 2023 to August 2023. The data summary is given in Table 3. For each subject, two images, one for each eye, were captured.

The analysis of the implemented solution based on the retinal parameter approach resulted in the understanding that artery-related measurements, namely CRAE, mean arteriole width, and AVR, appear to be more significant in distinguishing between cardio and healthy fundus images for the clinical trials conducted. This correlates to the following clinical observations in cardio cases like retinal arteriolar narrowing, a decrease in the arteriovenous ratio, and reduction in arteriole width. The measure of these distinguishing retinal parameters is depicted in the graph in Figure 4. The *x*-axis in this figure indicates the number of cases and the *y*-axis indicates the parameter values for cardio and healthy cases, separately. From Figure 4, it can be observed that a clear band is established in the *y*-axis (parameters) between healthy and cardio subjects. Any subject falling in the lower part of the cardio band can be considered as risk for cardio conditions and should take preventive measures to avoid further complications.

The DL-based binary classification model was tested and validated on 33 images for both categories, and the trained model resulted in accuracy, sensitivity, specificity, and F1 scores of 87.5, 85.29, 86.36, and 86.15, respectively. The confusion matrix of the binary classification model validation is given in Table 4.

Heatmap images were generated from the model prediction score to understand the relevant areas in the retinal image and the patterns of interest to the AI model to perform the classification. This acts like adding an explainability to the model prediction scores.

It is beneficial to visualize what the convolutional neural network values when it does a prediction. This allows us to see whether the model is on track, as well as what features it finds important. To visualize the heatmap, a technique called Grad-CAM (gradient class activation map) is used [36]. “Grad-CAM uses the gradients of any target concept (cardio or non-cardio), flowing into the final convolutional layer to produce a coarse localization map highlighting the important regions in the image for predicting the concept”.

To find the importance of a certain class in the AI model, we take its gradient with respect to the final convolutional layer and then weigh it against the output of this layer. The output of Grad-CAM is a heatmap visualization for a given class label. Heatmap images and corresponding retinal images are shown in Figure 5. A scale is given below the heatmap that indicates the values of the colors on the heatmap. Grad-CAM works by finding the final convolutional layer in the network and then examining the gradient information flowing into that layer.

The model focuses mostly on the center of the retinal images where signs of CVD are prevalent. This can be observed from the heatmap images. This indicates that our classification model can identify regions of certain prognosis markers for hypertension and ischemic heart disease. The prognosis markers can be utilized by the experts to assess the likelihood of disease progression, tailor treatment strategies according to individual risk profiles, monitor patients for early signs of complications, and ultimately improve patient outcomes through personalized care and intervention.

## 4. Discussion

The retinal images and metadata collected via the clinical trials conducted at the Ramesh Cardiology clinic aided in validating the implemented parameter-based approach for CVD risk assessment. The metadata collected served the purpose of curating images. These images were used to train the deep-learning-based cardiovascular risk classification model and to validate the accuracy of parameter-based risk assessment. However, the execution of the trained model and the implemented approach mandates only the provision of retinal images as input. Analysis of all collected images facilitated examination of key retinal parameters, revealing distinct differences among CRAE, AVR, and mean arteriole width.

The categorized images based on the metadata were used to develop a binary classification DL model that would complement the risk stratification obtained from the parameter-based analysis. Once both the approaches were found to be stable individually, these two approaches were integrated, and the efficacy of the combined risk analysis was computed. The analysis of the combined risk analysis is given below in Table 5. A total of 44 images from the two categories (risk for cardio and no-risk for cardio) were used for validating this combined risk analysis. The combined risk analysis proved to have an accuracy of 85 percent. Table 5 gives the confusion matrix for the combined risk analysis.

### 4.1. Implications of the Hybrid Approach

The primary distinguishable contributions of this research are delineated as follows:Customized Solution for CVD Risk Evaluation: This work presents a bespoke approach for assessing the risk of CVD. By customizing the methodology, it aims to provide a more precise assessment of cardiovascular health.Hybrid Architectural Framework: Introducing a hybrid architecture, this research combines multiple methodologies leveraging both handcrafted features and AI-driven retinal vascular pattern analysis. This amalgamation enhances the robustness and efficacy of the proposed system.Superior Performance in Clinical Trial Validation: Through meticulous clinical trial validation, this research establishes the superiority of its methodology over existing systems. By showcasing improved performance metrics, it underscores the potential clinical utility and reliability of the developed approach.

The hybrid approach presented in this study holds significant implications for CVD risk assessment as it integrates computer vision techniques with AI models. The proposed methodology offers a novel and efficient way to evaluate CVD risk using retinal fundus images. This approach can revolutionize current practices by providing an accessible method for early detection and monitoring of cardiovascular health. Furthermore, the proposed hybrid architecture enhances the accuracy and reliability of CVD risk prediction.

### 4.2. Comparison with Existing Solutions in CVD Risk Assessment

The results of our study, particularly regarding the retinal parameters CRAE, AVR, and mean arteriole width, are consistent with the existing literature on CVD risk assessment using retinal imaging. In our study, we observed distinct differences in these parameters between individuals classified as high risk and those classified as low risk for CVDs. Specifically, we found that individuals at high risk exhibited narrower CRAE, lower AVR values, and decreased mean arteriole width compared to those at low risk. These findings align with previous studies investigating retinal parameters as biomarkers for cardiovascular health. Several similar studies have reported analogous trends, wherein individuals with narrower retinal arterioles, lower AVR, and altered arteriole width are more likely to have an elevated risk of developing cardiovascular diseases. This consistency across studies underscores the robustness of retinal parameters as indicators of cardiovascular risk.

Our study contributes to the advancement of CVD risk assessment by addressing several limitations of the existing solutions. Unlike most traditional methods that rely solely on retinal parameter computation and analysis, our hybrid approach combines computer vision with AI algorithms to extract meaningful insights from retinal images. This integration enables a more comprehensive analysis of vascular patterns and microvascular changes associated with CVDs, leading to improved diagnostic accuracy and risk prediction. The hybrid risk assessment ensures the minimization of false positive cases and an increase in the true positive cases. This has been established in the validation phase of the clinical trials.

## 5. Conclusions

This research demonstrates the potential of leveraging retinal fundus images for cardiovascular risk assessment, offering a non-invasive and accessible approach to early detection and intervention. The integration of advanced AI techniques and computer vision algorithms allows for the extraction of valuable vascular features, providing a comprehensive understanding of the intricate relationship between retinal microvascular changes and systemic cardiovascular health. The presented hybrid approach exhibits promising results in predicting key cardiovascular risk factors, showcasing its ability to contribute to personalized risk stratification. The interpretability of the approach, enhanced through visualization, facilitates a better understanding of the underlying mechanisms contributing to risk predictions. The study highlights the potential for fundus image analysis to serve as a scalable solution for population-wide screening programs. The non-invasiveness and cost-effectiveness of this modality make it particularly appealing for large-scale healthcare initiatives aimed at preventing and managing cardiovascular diseases. By identifying vascular abnormalities before the onset of overt cardiovascular events, the approach aligns with the principles of preventive medicine and offers a proactive strategy for cardiovascular health management. This research presents a significant step forward, and ongoing efforts will focus on refining the AI model’s robustness, exploring additional risk factors, and validating its performance across diverse clinical settings. The ultimate goal is to integrate this innovative tool into routine clinical practice, improving patient outcomes in the realm of cardiovascular health.

### Challenges and Future Directions

Retinal-based CVD risk assessment, being a cross-domain analysis, poses a challenge with establishing equivalence between retinal parameters and CVD risk standard measurements. Further research is required to bridge this gap and improve acceptability by both patients and experts. Standardizing imaging protocols, interpreting findings, and integrating retinal imaging into routine clinical practice pose significant challenges. Yet, advancements in imaging technology, data analytics, and collaborative efforts between ophthalmology and cardiology hold promise for overcoming these hurdles. These ongoing developments offer opportunities for the further exploration and refinement of this approach, ultimately enhancing its effectiveness in assessing cardiovascular health and guiding clinical decision-making.

In summary, leveraging the retina as a window into cardiovascular health through specialized imaging techniques offers a promising avenue for non-invasive early detection and monitoring of cardiovascular risk factors. This can potentially lead to more proactive and personalized interventions to prevent cardiovascular diseases. Retina-based cardiovascular assessment holds promise in clinical practice as a complementary method for evaluating cardiovascular health. It enables the early detection and monitoring of cardiovascular risk factors, aiding in risk stratification and guiding preventive interventions. Integrating retinal examination into routine health screenings could enhance cardiovascular risk assessment and facilitate timely interventions to mitigate potential health risks.

## Figures and Tables

**Figure 1 diagnostics-14-00928-f001:**
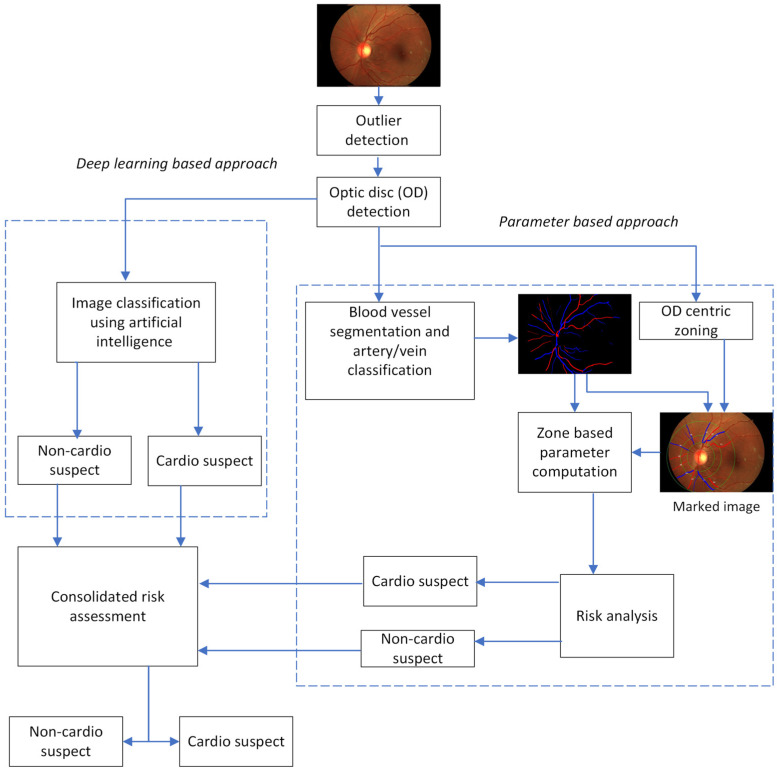
High-level architecture of the proposed retina-based consolidated CVD assessment system.

**Figure 2 diagnostics-14-00928-f002:**
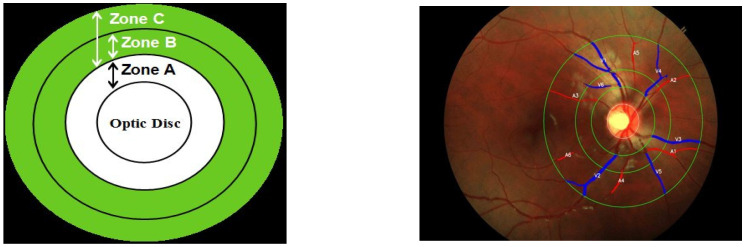
Retinal zones and AV classification.

**Figure 3 diagnostics-14-00928-f003:**
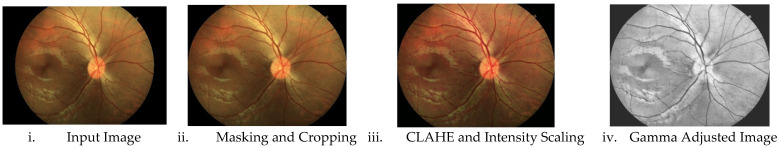
Phases of pre-processing applied for the binary classification model.

**Figure 4 diagnostics-14-00928-f004:**
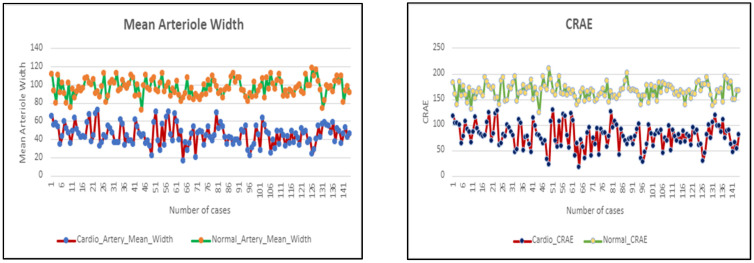
Observed parameter trends (cardio vs. healthy).

**Figure 5 diagnostics-14-00928-f005:**
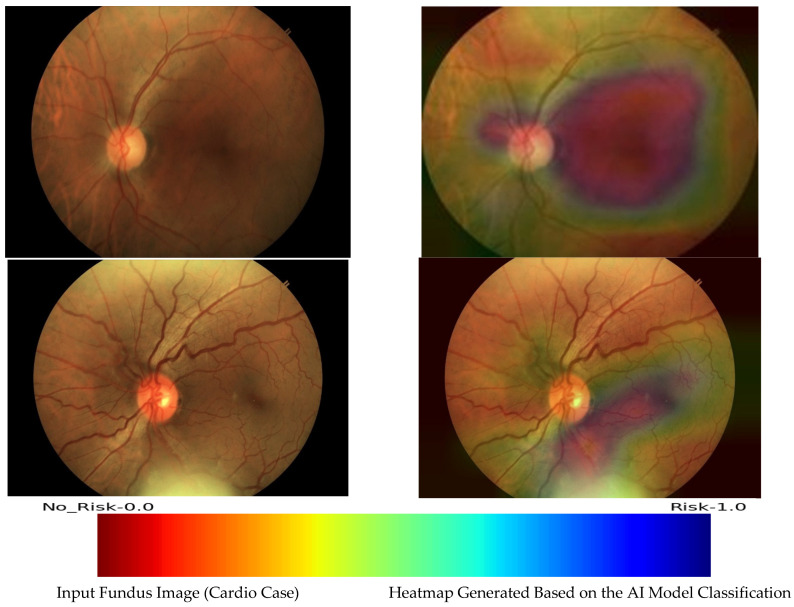
Heatmap images depicting the relevant pattern for the model.

**Table 1 diagnostics-14-00928-t001:** Dataset split for outlier detection model.

Classes	Training (Number of Images)	Validation (Number of Images)
Non-fundus and poor quality fundus images	904	300
Fundus image	956	300

**Table 2 diagnostics-14-00928-t002:** Dataset split for binary classification model.

Classes	Training (Number of Images—After Augmentation)	Validation (Number of Images—Original)
Cardio cases	1500	33
Non-cardio cases	1500	33

**Table 3 diagnostics-14-00928-t003:** Clinical data summary.

Condition	Total Screenings (No. of Subjects)
Hypertension ^~^	145
Cardio specific ^#^	234
Healthy *	140

* Healthy—The subjects had no concomitant medical or neurological conditions like hypertension, diabetes, or stroke and no history of eye trauma. ^#^ Cardio specific—The subjects considered for study are under treatment for cardiovascular conditions. ^~^ Hypertension—The subjects considered for study are under treatment for hypertension.

**Table 4 diagnostics-14-00928-t004:** Confusion matrix for binary classification model.

Actual			
Predicted		Cardio	Healthy
	Cardio	28	5
	Healthy	4	29

**Table 5 diagnostics-14-00928-t005:** Confusion matrix for combined risk analysis.

Actual			
Predicted		Risk	No-Risk
	Risk	40	4
	No-Risk	9	35

## Data Availability

Data are unavailable due to privacy or ethical restrictions.

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
