# Peer review of "A Multi-Stage Approach for Cardiovascular Risk Assessment from Retinal Images Using an Amalgamation of Deep Learning and Computer Vision Techniques"

_diagnostics, 2024, doi:10.3390/diagnostics14090928_

Round 1
Reviewer 1 Report
Comments and Suggestions for Authors
Main concerns:
The study lacks informations regarding demographics and exact disease characteristics (how many with angina, IHD etc). It is not clear if 300 patients were enrolled as stated in Design, or more as stated in Table 3.
Lines 91-92 could imply that CSLO or OCT will be used in the study, which was not the case.
The study implies 2 methods to assess risks :parameter-based analysis and deep learning techniques. However, in the results there is no clear data from both, Table 4 states that only 33 patients for deep learning were used, there is no comparison from the 2 methods.
The study lacks any discussion. There are many study assesing the same risks, e.g doi: 10.1161/CIRCULATIONAHA.121.057709 (a study on 97 895 retinal fundus image). What is new in the present study?
Lines 142-175 should be eliminated. The topic is allready covered in the introduction. Embriological development does not aid in understanding the article.
Line 184- the information about retina keeps repeting (`the part of eye that receives,...early indicator of circulatory disease`). All similar issues regarding repetition of information should be adressed (e.g lines 202-204, but many others)
Line 226 - missing word ~It~
It is better to move the Literature review at line 112. The purpose of the present article will follow the review and beter introduce the study design.
Lines 287 and 302 repeats information, should be condensed
Table 3 -please explain symbols in the table #,* etc
Author Response
Dear Reviewer,
We would like to express our sincere gratitude for your valuable feedback and constructive comments on our manuscript titled “A Multi-stage Approach for Cardiovascular Risk Assessment from Retinal Images using an Amalgamation of Deep Learning and Computer Vision Techniques “. We have carefully considered all the review comments and suggestions given by the you and have incorporated the same in the revised manuscript. Your suggestions have greatly contributed to the improvement of the paper's content quality, and we express sincere gratitude for the time and effort you have dedicated to reviewing our submission.

Reviewer 2 Report
Comments and Suggestions for Authors
It is an intriguing study. The functional and morphological changes of the retina have been increasingly valued as non-invasive biomarkers for various systemic diseases, including cardiovascular diseases (CVD) and central nervous system diseases, among others. We should fully leverage the advantages of AI technology to achieve these goals. Beyond merely detecting the presence of disease, it would be more exciting to see models that can help predict future occurrences and prognoses. The design of combining two methods to double check the output is novel.
Major Concerns:
1. It appears that there is no description of how the dataset was divided for the training, validation, and testing processes.
2. It was stated that “… This indicates that our model can identify and make use of certain prognosis markers for hypertension and ischemic heart disease.” I understand that this procedure can detect and evaluate CVD, can you explain why it can be used as prognosis markers?
3. Should a heatmap include a scale so that it can more clearly display what specific values the different colors on the heatmap represent, thereby providing a more precise interpretation of the information?
4. How does the metadata contribute to the model?
5. Can you further discuss the difference or the consistence of your results with other similar studies?
Minor comments:
1. Font is inconsistent throughout the manuscript, please check.
2. Introduction section line 91, it should be “…optical coherence tomography angiography (OCT)… are used to capture high-resolution images of the retinal vessels.”
3. Introduction section lines 78-111 contain too much repeated information, and the logic is unclear; please rephrase.
4. The content in lines 112-140 of the Introduction section resembles paragraphs from a funding proposal rather than a scientific article. Please restructure this to formulate a paragraph that clearly states the main aim of this study for the Introduction. The approaches should be detailed in the Methods section, while the main contributions ought to be addressed in the Discussion section.
5. Could you provide more detailed descriptions for each figure instead of just a sentence? Also, some of the figures need to be renumbered.
6. Table 1-2 and figure 3 can be included as supplementary documents.
Author Response

(The authors gave the same response as above.)

Round 2
Reviewer 1 Report
Comments and Suggestions for Authors
no further suggestion